# The Role of Choline in Neurodevelopmental Disorders—A Narrative Review Focusing on ASC, ADHD and Dyslexia

**DOI:** 10.3390/nu15132876

**Published:** 2023-06-25

**Authors:** Emma Derbyshire, Michael Maes

**Affiliations:** 1Nutritional Insight, Surrey KT17 2AA, UK; 2Department of Psychiatry, Faculty of Medicine, Chulalongkorn University, Bangkok 4002, Thailand; 3Research Institute, Medical University of Plovdiv, 10330 Plovdiv, Bulgaria

**Keywords:** attention deficit hyperactivity disorder, autism spectrum condition, autism spectrum disorder, brain function, choline, neurodevelopment, neurotransmitters, pregnancy, specific learning disorders

## Abstract

Neurodevelopmental disorders appear to be rising in prevalence, according to the recent Global Burden of Disease Study. This rise is likely to be multi-factorial, but the role of certain nutrients known to facilitate neurodevelopment should be considered. One possible contributing factor could be attributed to deficits in choline intake, particularly during key stages of neurodevelopment, which includes the first 1000 days of life and childhood. Choline, a key micronutrient, is crucial for optimal neurodevelopment and brain functioning of offspring. The present narrative review discusses the main research, describing the effect of choline in neurodevelopmental disorders, to better understand its role in the etiology and management of these disorders. In terms of findings, low choline intakes and reduced or altered choline status have been reported in relevant population subgroups: pregnancy (in utero), children with autism spectrum disorders, people with attention deficit hyperactivity disorder and those with dyslexia. In conclusion, an optimal choline provision may offer some neuronal protection in early life and help to mitigate some cognitive effects in later life attributed to neurodevelopmental conditions. Research indicates that choline may act as a modifiable risk factor for certain neurodevelopmental conditions. Ongoing research is needed to unravel the mechanisms and explanations.

## 1. Introduction

Neurodevelopmental disorders (NDDs) are a class of disorders impacting brain development and function [1]. In the Diagnostic and Statistical Manual of Mental Disorders, Fifth Edition (DSM-5) NDDs are defined as a group of conditions with onset in the developmental period, inducing deficits that produce impairments of functioning [2]. Within this definition, NDDs consist of: autism spectrum disorder (ASD; a communication disorder); attention-deficit/hyperactivity disorder (ADHD); intellectual disabilities; neurodevelopmental motor disorders (including tic disorders) and specific learning disorders (including dyslexia) [2,3]. A high level of comorbidity also exists between conditions. For example, ASD and ADHD are known to have shared genetic heritability, with both being associated with social and executive functioning impairments [4]. Most individuals with ASD exhibit ADHD symptoms, and around 15–25% of ADHD individuals have ASD symptoms [5]. Similarly, research from twin studies found that there was more than an eightfold increase in the prevalence of NDDs (termed ‘neurodevelopmental disorders and problems’ in this study) in individuals with dyslexia, compared with typical readers [6].

The global burden of NDDs appears to be rising, as demonstrated in Figure 1. Both ASD and ADHD are reported to have risen in prevalence over the past 10 years [5]. An analysis using data from 204 countries and territories forming part of the Global Burden of Disease Study 2019 showed that for ASD, age-standardized rates had risen by around 0.06% annually over the last three decades [7]. The total global prevalence of ASD in 1990 was 20.3 million, increasing to 28.3 million in 2019 [8]. For males, ASD prevalence was 15.6 million in 1990 and 21.6 million in 2019 [8]. For females, the prevalence of ASD in 1990 was 4.7 million, rising by an additional 2 million to 6.7 million by 2019 [8]. For ADHD, in 1990 the reported prevalence was 72.4 million, rising to 84.7 million in 2019 [8]. In males, the global prevalence of ADHD was higher—52.6 million in 1990, increasing to 61.5 million in 2019 [8]. Amongst females, the 1990 global prevalence of ADHD was 19.8 million, rising to 23.2 million in 2019 [8]. Subsequently, the burdens of ADHD and ASD appear to have been greater in males than females [8]. ADHD and ASD remain under-recognized and underdiagnosed in many countries, especially amongst girls and women [9,10]; thus, prevalence rates could be even higher. Prevalence is also reported to be higher in certain population groups, such as looked-after children [11,12]. ADHD prevalence transitions into adulthood in around 30–50% of cases [13].

Dyslexia is highly prevalent, affecting around 20% (1 in 5) of the global population, and males/females equally [14]. It occurs across a range of cognitive and language abilities, a range which includes both higher-than- and lower-than-average levels of functioning [15]. There are different definitions of dyslexia, but Reid (2016) defines it succinctly as a “processing difference, often characterized by difficulties in literacy acquisition affecting reading, writing, and spelling. It can also have an impact on cognitive processes such as memory, speed of processing, time management, coordination, and automaticity. There may be visual and/or phonological discrepancies and there are usually some discrepancies in educational performances” [16].

NDDs are now the most frequently diagnosed conditions in child neurology/pediatric clinical practices [17,18]. There are many potential explanations as to why the prevalence of NDDs could be rising. Improved diagnostic screening might be one explanation [18]. Other factors such as maternal metabolic conditions [19] and misalignment with dietary and lifestyle recommendations have also been proposed [20]. Increasingly, the role of nutrition during gestation (pregnancy) and neurodevelopment is increasingly being recognized, with inadequate intakes of certain nutrients being linked to ADHD, ASD, altered cognition and visual and motor deficits [21]. Past research has focused heavily on nutrients such as the omega-3 fatty acid docosahexaenoic acid (DHA) [22,23,24], but now research has accrued for other nutrients, including choline, considering their roles in neurodevelopment and promoting optimal cognition [21,25,26,27,28].

In this narrative review, we focus on and examine the role(s) of choline as a potential modifiable risk factor for certain NDDs. We will focus on ADHD, ASD (now also referred to as ASC, autistic spectrum condition), and dyslexia, as this is where most research appears to sit.

## 2. Neurodevelopment

The brain is a central organ that orchestrates the whole body [29]. The human brain begins to develop as early as the third week into gestation when neural progenitor cells differentiate; this process extends into later adolescence and potentially across the lifespan [30]. Processes underpinning brain development include gene expression and environmental inputs, which are both crucial for normal brain development, with disruption of either significantly impacting upon neural outcomes [30]. The development of the brain’s circuitry begins as early as 2–3 weeks into gestation and requires the coordination of complex neurodevelopmental processes [31]. Stiles et al. (2010) explains that “brain development is aptly characterized as a complex series of dynamic and adaptive processes that operate throughout the course of development to promote the emergence and differentiation of new neural structures and functions” [30].

From approximately 6 weeks post-conception to mid-gestation, a number of cellular events occur, including neurogenesis followed by apoptosis, differentiation, migration and synapse formation [31]. Extended periods of cortical development occur across the lifespan. This ‘developmental’ phase occurs across childhood and adolescence when cortical development (outer layers of the cerebrum) transitions from lower-order, unimodal cortices with motor and sensory functions to higher-order, trans-modal cortices underpinning executive, socioemotional and mental brain functions [32].

The role of in utero programming, as described in the theory developed by Professor David Barker, and thus coined the ‘Barker Hypothesis’, is well-recognized [33]. If fetuses have a limited nutrient supply in utero they need to adapt; this is a process that can permanently modify their structure/metabolism, inducing ‘programme changes’ that can be the origins of other conditions later in life (the ‘fetal origins of adult disease’) [33]. It is now well recognized that changes in brain function can lead to a spectrum of NDDs [29].

Intrauterine exposures (including nutrients) have been linked to NDDs, although elucidating the timing and exact mechanisms can be challenging [34,35]. Such exposures during these sensitive windows of life are another factor considered to influence brain development [36]. Increasingly, it is well appreciated that normal neurodevelopment is central for brain functions across the lifespan, with any modulations potentially contributing to brain dysfunction [29]. Heland et al. (2022) recently proposed that nutritional deficits, to some extent, could potentially prevent neurodiversity, as certain nutrients have the ability to improve neurodevelopmental outcomes by mitigating pathological processes such as inflammation, hypoxia and oxidative stress [37]. This brings us on to the potential role of choline.

## 3. In Utero Origins

Zeisel et al. (2006) described how choline deficiency during sensitive periods of brain development could induce permanent changes in brain function and result in persistent cognitive and memory deficits [38]. As choline is important for brain development, Bernhard et al. (2013) raised concerns about choline levels in very low birth-weight infants, with nutritional intakes of preterm infants frequently being less than the estimated adequate intake, and shortages being apparent until day 10 postnatally [39].

In the Seychelles Child Development Nutrition Study, choline was listed as one of the key nutrients expected to have direct effects on neurodevelopment, both prenatally and postnatally, and was believed to have some correlation with fish consumption [40]. Another study looking at maternal egg consumption found that choline deficiency predicted fetal autonomic and brain maturation indices at 32- and 36-weeks’ gestation, respectively [41]. Poor availability of choline in utero has been further linked to impaired differentiation of retinal neuronal cells, indicating a role in the development of the visual system [42].

Derbyshire and Obeid (2020) [27] provided an updated systematic review using data from 38 animal and 16 human studies. In particular, it was concluded that choline over the first 1000 days of life could potentially: (1) support normal brain development; (2) protect against neural and metabolic insults, including alcohol; and (3) improve neural and cognitive functioning [27]. A further systematic review and meta-analysis collating evidence from 30 publications found that higher maternal choline intake was likely to be associated with improved child neurocognition/neurodevelopment [26].

### Gestation

Choline deficiency is common in pregnancy (in utero exposure) [43,44]. Average choline intakes amongst women of childbearing age have been explored in a review of 23 studies, and were reported to range from 233 mg/day to 383 mg/day, even with the inclusion of choline from supplements, and thus are consistently lower than the estimated adequate intake (AI) of 480 mg/day for pregnant women [45,46]. In a recent study conducted in Germany, only 7% of pregnant women achieved adequate choline intakes [47]. Similarly, amongst an Australian sample of pregnant women, median choline intake was 362 mg/day in early pregnancy, with eggs providing around 17% of the choline [48]. The authors concluded that few pregnant women met the AI for choline, and that this may need to be improved [48].

## 4. Mechanistic Studies

Choline is an essential micronutrient, as recognized by the United States (U.S.) Institute of Medicine in 1998 [49]. Mechanistically, it is a precursor of the brain neurotransmitter acetylcholine and membrane phospholipids, including phosphatidylcholine [50,51]. It is also a methyl donor known to play a central role in brain growth and development, maintaining the functional and structural integrity of the cell membrane [36]. Through the actions of its metabolites, it partakes in pathways involved in the methylation of genes related to memory and cognitive functions [36].

As shown in Table 1, several mechanistic studies have investigated the effects and mechanisms of choline on brain development. The fetus and newborn are known to have high choline demands, with the micronutrient’s role in DHA and histone methylation believed to be as a modifier of genes involved in aspects of learning and memory [52]. Bekdash et al. (2016) describe how choline is an important epigenetic modifier of the genome (altering gene methylation, expression and cell function), with abnormal levels during fetal development and/or early postnatal life being linked to altered memory functions later in adult life [53]. Choline is also needed for normal memory development, possibly due to changes in the development of the memory center (hippocampus) in the brain [54]. Choline is thought to influence stem cell proliferation and apoptosis, therefore potentially modifying brain structure and function [55].

Murine models show that postnatal choline treatment can modulate neuronal plasticity, preventing deficits in motor coordination whilst enhancing density of dendritic spines and neuronal morphology [56]. Choline has further been found to partially restore dendritic structural complexity in murine hippocampal neurons that are iron deficient [57]. Other work shows that reduced choline supplies during gestation can impede and diminish the number of cortical neural progenitor brain cells (NPCs), with two types of NPCs—the radial glial and intermediate progenitor cells—being affected, indicating that choline supplies regulate cerebral cortex development [58]. A murine model focusing on autism showed that choline supplementation administered to offspring of methylenetetrahydrofolate reductase (MTHFR)-deficient mothers had the potential to attenuate the autistic-like phenotype [59]. Further research found choline supplementation to improve impairments in social interaction in a murine model of autism, helping to reduce deficits in social behavior and reduce anxiety [60].

**Table 1 nutrients-15-02876-t001:** Role(s) of choline in neurodevelopment and brain function.

Author(s), Year	Publication	Choline Mechanisms
Blusztajn et al. (2017; 2012) [50,52]	Discussion paper	Precursor of PC, acetylcholine, and via betaine, the methyl group donor S-adenosylmethionine. Cho as a methyl donor influences DNA and histone methylation (regulates gene expression).
Bekdash et al. (2019; 2018; 2016) [36,53,61]	Discussion paper	Cho maintains structural and functional integrity of membranes and regulates cholinergic neurotransmission via acetylcholine synthesis.
Zeisel et al. (2020; 2017; 2011; 2006; 2000; 1997) [38,54,55,62,63,64]	Discussion paper	Cho can induce changes in the development of the memory center (hippocampus), modulate methylation via BHMT (and its metabolite, betaine) and regulate S-adenosylhomocysteine and S-adenosylmethionine levels. Plausible mechanisms: changes in transmembrane signal transduction, stem cell proliferation/differentiation, regulation of apoptosis, gene expression in neuronal and endothelial progenitor cells.
Bastian et al. (2022) [57]	Murine model	Cho restored dendritic function in developing hippocampal neurons that were iron-deficient.
Agam et al. (2020) [59]	MTHFR-deficient mice	Cho supplementation, even at adulthood, to offspring of MTHFR-deficient mothers attenuated the autistic-like phenotype.
Chin et al. (2019) [56]	Knockout murine model	Cho enhanced neuronal morphology, rescued deficits in motor coordination and increased density of dendritic spines.
Wang et al. (2016) [58]	Murine model	When Cho supply was reduced during gestation, the number of two types of cortical NPCs (radial glial cells and intermediate progenitor cells) were significantly reduced in fetal brains.
Langley et al. (2015) [60]	Mouse model of autism	Cho intake during early development can prevent/dramatically reduce deficits in social behavior and anxiety.

Key: BHMT, betaine homocysteine methyltransferase; Cho, choline; DNA, deoxyribonucleic acid; MTHFR, Methylenetetrahydrofolate reductase; NPCs, neural progenitor cells; PC, phosphatidylcholine.

## 5. Brain Imaging (Animal Models)

Several imaging studies have investigated choline’s role in brain structure and function using animal models. Mudd et al. (2016) fed sows either a choline-deficient (CD) or choline-sufficient (CS) diet for the last half of gestation, finding that CD sows had smaller total brain volumes when measured using magnetic resonance imaging 30 days postnatally [65]. Concentrations of glycerophosphocholine-phosphocholine were also lower, indicating that choline deficiency appeared to delay neurodevelopment and induce structural and metabolic changes [65]. Other work by Mudd et al. (2018), using a similar approach, further showed that CD pigs had significantly reduced left- and right-cortex gray matter compared with prenatally CS pigs [66].

## 6. Human Studies

### 6.1. ASD

Nutritional status (including low choline) is regarded as playing a key role in the severity and incidence of core ASD symptoms [67]. Research from human studies has focused on the potential role(s) of choline in relation to ASD (Table 2).

Firstly, work using spectroscopic imaging on children with ASD (aged 3–4 years) found that gray matter and white matter levels of choline were reduced, compared with typically developed children [68,69]. Focusing on the white matter composition, other work also found that ASD and its severity was associated with lower levels of choline in brain white matter, along with the perisylvian cortex [70]. Similarly, Margari et al. (2018) found brain metabolite levels (choline/Cr ratios) to be significantly altered in the frontal lobe white matter in ASD subjects versus controls [71]. Hardan et al. (2008) observed lower levels of choline in the left side of the thalamus in children with autism [72]. Some research has found lower choline levels in the thalamus to be correlated with behavioral scores in ASD children (7–18 years), i.e., increased severity of stereotyped behaviors and communication impairments [73].

Regarding metabolic characteristics, Wang et al. (2022) studied 29 ASD and 30 typically developing boys (mean age ≈ 3 years) [74]. Boys with ASD had lower levels of plasma choline, which was adversely correlated with ABC-language scores–findings that aligned with Gabis et al. (2019) [74,75]. Indeed, the work by Wang et al. (2022) also found that choline metabolism intermediates such as phosphatidylcholine and lysophosphatidylcholine (involved in glycerophospholipid metabolism) were reduced, implying that this could impact on processes of choline metabolism and subsequent impairments in language ability in ASD children [74]. Additional work has investigated the effects of supplementation. Gabis et al. (2019) undertook a double-blind randomized trial examining the combined effects of donepezil and choline (350 mg/day for 8 weeks in the open label study phase) compared to a placebo [75]. The treatment group had a sustained effect on receptive language skills in ASD children for 6 months post-treatment, with more significant effects observed in those under 10 years of age [75].

Other research has described habitual choline intake and status. Scientists using data from the U.S. Autism Intervention Research Network for Physical Health (AIR-P) study found that 60–93% of ASD children were consuming less than the recommended adequate intake for choline [76]. Children with autism also had lower plasma choline levels than did healthy controls [76]. The authors concluded that choline intakes were inadequate in a significant number of ASD children, which could contribute to abnormalities in folate-dependent one-carbon metabolism observed in many children with autism [76]. Very similar findings were reported by Hyman et al. (2012) [77]. An analysis of 3-day food records from children with ASD aged 2–11 years showed that few ASD or matched controls met recommended chorine intake levels [77].

**Table 2 nutrients-15-02876-t002:** Key studies investigating ASD and choline.

Author, Year	Study Design/Approach	Age	No. of Participants	Key Findings
Wang et al. (2022) [74]	Metabolomic analysis	ASD 3.02 ± 0.67 y TD 3.13 ± 0.46 y	*n* = 28 boys ASD *n* = 30 boys TD	The level of Cho was inversely correlated with ABC-language score in ASD group.
O’Neill et al. (2020) [70]	Magnetic resonance spectroscopy	5–60 y	*n* = 78 ASD *n* = 96 TD	Cho metabolites were lower in ASD than in TD.
Gabis et al. (2019) [75]	Randomized, DB, PC trial (12-wk int., 6-months washout)	NR	*n* = 60 ASD	Donepezil hydrochloride + Cho contributed to a sustainable effect on receptive language skills in ASD children for 6 months after treatment, with a more significant effect in those <10 y.
Margari et al. (2018) [71]	Case-control	21 months to 14 y, 1 month	*n* = 75 ASD *n* = 50 controls	Cho/Cr ratios were significantly altered in ASD vs. controls.
Doyle-Thomas et al. (2014) [73]	Case-control	7–18 y	*n* = ASD *n* = 16 TD controls	Boys had increased Cho in the thalamus. Cho in the three brain regions studies correlated with behavioral scores in the ASD group
Corrigan et al. (2013) [69]	Cross-sectional analysis	3–4 y 7–9 y 9–10 y	*n* = 45 ASD *n* = 14 DD *n* = 14 TD	At 3–4 y, the ASD group exhibited lower Cho concentrations than did the TD group
Hamlin et al. (2013) [76]	Observational analysis	2–11 y	*n* = 288 ASD	Cho intake was inadequate in a significant subgroup of ASD children and reflected in lower plasma levels. This may contribute to metabolic abnormalities.
Hyman et al. (2012) [77]	Prospective analysis	2–11 y	*n* = 252 ASD	Few children met the recommended intakes for Cho.
Hardan et al. (2008) [72]	Case-control	8–15 y	*n* = 18 M ASD *n* = 16 controls	Lower levels of Cho-containing metabolites were found on the left side of the thalamus in the ASD group vs. controls.
Friedman et al. (2006) [68]	Cross-sectional spectroscopic imaging	3–4 y	*n* = 45 ASD *n* = 12 DD *n* = 10 TD	Children with ASD had significantly decreased gray matter concentrations of Cho (*p* < 0.001)

Key: ASD, autism spectrum disorder; Cho, choline; DB, double-blind; DD, delayed development; int., intervention; M, male; PC, placebo-controlled; TD, typical development; y, years.

### 6.2. ADHD

Some human trials have included an analysis of choline status or circulating levels of brain metabolites and studied these factors in relation to ADHD (Table 3).

Focusing on imaging studies, Alger et al. (2021) used whole-brain diffusion tensor imaging [78]. The results showed that ADHD children (aged 8–13 years) exposed to ‘prenatal alcohol exposure’ (PAE) had more severe white-matter pathology compared with those without PAE [78]. Amongst those with ADHD + PAE, Tower Test Achievement scores (higher for better performance) correlated negatively with choline, with its role somewhat unclear and warranting continued investigation [78]. Additional neuroimaging work measuring prefrontal white matter discovered that anterior corona radiata levels of choline were 27% lower in children and adolescents with ADHD + PAE, compared with those with idiopathic ADHD [79]. Other magnetic resonance research showed that 20–25% of neurons may have died or been severely dysfunctional in pediatric ADHD patients, and there seemed to be mild hyperactivity of the cholinergic system (the system that encompasses the synthesis and secretion of acetylcholine) [80].

Additional research has looked at supplementation. One randomized controlled trial provided 2–5-year-olds with fetal alcohol spectrum disorder (FASD) with choline (1.25 g. choline bitartrate powder delivering 513 mg/day choline) or a placebo over 9-months, with follow up 4 years later (mean age 8.6 years) [81]. At follow-up the children who had received choline had fewer behavioral symptoms of ADHD than did the placebo group [81]. The children administered choline also had better working memory ability, verbal memory, visual-spatial skills, and non-verbal intelligence than the placebo group [81]. Borlase et al. (2020) studied the effects of micronutrient treatment on brain neurometabolites, since these appeared to be altered in children with ADHD, particularly in the prefrontal cortex and striatum [82]. Children with ADHD (mean age 10.8 years, non-medicated, *n* = 27) received daily micronutrients or a placebo over 10 weeks [82]. It was not specified which micronutrients or dosages were involved, but choline was identified as a metabolite of interest. In the treatment group, there was a trend for decreased (improved) choline in the left and right striatum, though changes were not regarded to be of significance [82].

Some research has studied choline levels in relation to methylphenidate administration. Earlier work proposed that choline does not appear to be sensitive to methylphenidate treatment in children [83]. However, an analysis of regional brain spectra in 2008 found a significantly reduced signal of choline-containing compounds following methylphenidate treatment [84]. This is believed to fit with a recent energetics hypothesis of ADHD—that insufficient lactate supply to oligodendrocytes leads to impairments in fatty acid synthesis and myelin sheath formation, which may account for the reduced choline levels [84,85]. According to Wiguna et al. (2012), stimulant (long-acting methylphenidate) treatment over 12 weeks appears to induce neurochemical changes (thought to reflect improved neuroplasticity) in the prefrontal cortices of children [86]. In particular, the choline/creatine ratio decreased significantly in children (mean age 8.5 years) by 12.4% in the right prefrontal cortex and 16% in the left prefrontal cortex [86]. This was a pilot study, so repeated investigation is needed.

**Table 3 nutrients-15-02876-t003:** Key studies investigating ADHD and choline.

Author, Year	Study Design/Approach	Age	No. of Participants	Key Findings
Alger et al. (2021) [78]	Case-control	8–13 y	*n* = 23 ADHD + PAE *n* = 19 ADHD − PAE *n* = 28 TD	Cho findings were less prominent in this study.
Borlase et al. (2020) [82]	Randomized PC trial	11 y (mean)	*n* = 27 ADHD (non-medicated)	In the treatment group (12 capsules/day; dose NS) there was a trend for decreased choline in the striatum.
Wozniak et al. (2020) [81]	Randomized, DB, PC trial (9-months)	2–5 y	*n* = 15 FASD Cho *n* = 16 FASD placebo	Children receiving Cho (1.25 g. Cho bitartrate powder delivering 513 mg/day Cho) had higher non-verbal intelligence, higher visual-spatial skill, higher working memory ability, better verbal memory and fewer behavioral symptoms of ADHD than those receiving the placebo.
O’Neill et al. (2019) [79]	Case-control	7–17 y	*n* = 44, by subgroup	Low Cho may stem from abnormal Cho metabolism.
Wiguna et al. (2012) [86]	Prospective analysis	9 y (mean)	*n* = 21 ADHD	The Cho/creatine ratio decreased 12% in the right prefrontal cortex after stimulant treatment.
Kronenberg et al. (2008) [84]	Spectroscopic test-retest study	Adults	*n* = 7 ADHD	Analysis of regional brain showed a significantly decreased signal of Cho-containing compounds following treatment with MPH.
Carrey et al. (2003) [83]	Magnetic resonance spectroscopy	NR	*n* = 14 ADHD	Cho metabolites did not demonstrate any change in response to medication.
Jin et al. (2001) [80]	Magnetic resonance spectroscopy	NR	*n* = 12 B ADHD *n* = 10 controls	In ADHD children the bilateral striatum Cho/Cr ratio showed a mild unilateral increase. There appears to be mild hyperactivity of the cholinergic system.

Key: B, boys; Cho, choline; DB, double-blind; FASD, fetal alcohol spectrum disorder; MPH, methylphenidate; NR, not reported; NS, Not clearly specified; PC, placebo-controlled; TD, typical development; y, years.

### 6.3. Dyslexia

An emerging body of evidence has looked at choline levels/metabolites in relation to markers of dyslexia (Table 4). In one study, reading ability and executive function were measured in 24 children (8 to 12 years) with dyslexia and 30 typical readers [87]. For females with dyslexia there was a strong, statistically significant inverse correlation between processing speed and choline [87]. It is well appreciated that individuals with dyslexia have prolonged response times, and metabolite changes appear to be present which could hold promise as possible markers for dyslexia [87]. This is one of the first studies to identify metabolite changes in regions not regarded as being part of the ‘classic’ reading circuitry.

Earlier work has shown that choline levels were negatively correlated with reading and related linguistic measures in phonology and vocabulary (i.e., higher concentrations were associated with reduced performance) in the occipital lobe [88,89], left temporoparietal region [90] and the cerebellum [91]. It seems tenable that higher choline levels in reading-related regions for those with reading difficulties could be indicative of high membrane turnover, white matter, and cellular density [87]. This aligns with evidence that myelination is impaired in this population, particularly in the white-matter tracts that pass the temporoparietal regions and the occipital lobe [92,93].

Other research focusing on the visual and temporo-parietal cortex showed that children with dyslexia, compared to controls, had choline levels that were 7.6% lower in the left temporo-parietal region and 5.5% lower in the visual cortex [94]. Amongst children, the higher the choline concentration, the faster the Rapid Automatized Naming (RAN), though this did not reach a level of significance [94].

**Table 4 nutrients-15-02876-t004:** Key studies investigating markers of dyslexia.

Author, Year	Study Design/Approach	Age	No. of Participants	Key Findings
Kossowski et al. (2019) [94]	MEGA-PRESS single-voxel spectroscopy	30.28 ± 4.09 D 28.02 ± 3.40 C 10.90 ± 0.98 D 11.21 ± 0.95 C	*n* = 36 adults (50% D) *n* = 52 children (50% D)	Adults vs. children were characterized by higher Cho in the left temporo-parietal and occipital cortices.
Horowitz-Kraus et al. (2018) [87]	Magnetic resonance spectroscopy	8–12 y	*n* = 24 dyslexia *n* = 30 TR	After adjustment for multiple comparisons, F with dyslexia showed strong significant inverse correlations between processing speed and Cho (r = −0.54, *p* = 0.005) levels.
Del Tufo et al. (2018) [89]	Magnetic resonance spectroscopy	8.1 y (mean)	*n* = 231, Metabolites measured in *n* = 70	There was an inverse association between Cho and reading ability.
Pugh et al. (2014) [88]	Longitudinal analysis	6.1–10.1 y	*n* = 75 reading abilities across a continuum (including those with RD)	Cho levels were inversely correlated with reading, phonology and vocabulary (possible links to white matter anomalies and hyperexcitability).
Bruno et al. (2013) [90]	Magnetic resonance spectroscopy	18–30 y	*n* = 31 young adults	Lower scores on phonological decoding and sight word reading measures were associated with higher Cho concentrations.
Laycock et al. (2008) [91]	Whole-brain volumetric MRI	NR	*n* = 10 M dyslexic *n* = 11 M controls	The dyslexic group had a lower ratio of NAA/Cho in the right cerebellar hemisphere and a higher ratio of Cho/Cr in the left cerebellar hemisphere, possibly indicative of excessive connectivity or abnormal myelination.

Key: C, Control; Cho, choline; D, Individuals with dyslexia; M, male; MRI, Magnetic resonance imaging; NAA, *N*-acetyl aspartate; NR, not reported; RD, reading difficulties (termed disability in cited publication); TR, typical readers; y, years.

### 6.4. Processing Speed and Attention

It is well recognized that processing speed and working memory can be reduced in both individuals with ADHD and those with dyslexia, with individuals having both conditions concurrently being most affected [95]. Caudill et al. (2018) examined the effects of maternal choline supplementation during pregnancy [96]. Women moving into their third trimester were randomly assigned to groups consuming 480 mg choline/d (*n* = 13) or 930 mg choline/d (*n* = 13) until delivery [96]. Infant information processing speed and visuospatial memory were studied at 4, 7, 10 and 13 months of age [96]. Intriguingly, mean reaction time (averaged across the four ages) was significantly quicker for infants born to mothers who took the higher levels of choline—930 (vs. 480) mg choline/d [96]. These findings suggest that maternal consumption of around twice the recommended amount of choline during the last trimester of pregnancy improves infant information processing speed.

When these children were followed-up at 7 years of age, children who had been exposed to choline at a level of intake of 930 mg/d from the third trimester of pregnancy had improved levels of sustained attention (i.e., a significantly better ability to maintain correct signal detections) [97]. This may be attributed to them being able to sustain cholinergic activity in the prefrontal cortex of the brain, a region which regulates attentional control [97]. These findings are interesting, and have wider potential implications, including potential studies focusing on attention/inattention and choline status in individuals with ADHD and/or dyslexia.

## 7. Discussion

In the past, there has been a tendency to focus on nutrients such as long-chain omega-3 fatty acids in relation to their effects on neurodevelopment and NDDs [22,23,98,99,100]. However, in recent years there has been an accumulation of evidence highlighting the potential role(s) of choline [27,38,54,63].

Choline is recognized as having three distinct physiological roles: (1) the synthesis of neurotransmitter acetylcholine, (2) acting as a major methyl donor and (3) preserving the structural integrity and lipid-mediated signaling for cell membranes [44,101]. Choline has an important role in neurodevelopment, with normal concentrations enlarging cholinergic neurons in size and number in the medial septum [28]. There is now an accruing body of science demonstrating that choline is important for neurological development and brain function [21,27]. Regarding mechanisms, it is possible that the role(s) of choline are different depending on the life stage and form of the NDD. Mechanisms are complex and have potential to be multi-faceted. Modulations in white-matter pathology, impaired myelination, altered levels of brain metabolites and potential compensatory mechanisms have all been described in relation to choline markers [66,68,70,78,85]. Further research is now warranted to build on present insights and disentangle these.

For ASD, we have seen how lower plasma choline levels have been correlated with diminished language scores, indicating that there could be plausible links between products of choline metabolism and language ability [74,75]. Choline is a major brain metabolite and essential component of different membrane phospholipids [102]. Interestingly, other work conducted with typically developed children has discovered links between speeded (rapid) naming (a measure of long-term memory) and choline levels which could potentially influence language ability [103]. We have seen in several studies included in this narrative review how choline intakes and/or status appear to be reduced in children with NDDs, particularly ASD [74,76,77]. These are intriguing findings worthy of future development and research.

Moving on to ADHD, the effect of stimulant medication (methylphenidate) appears to be inconclusive, with some authors reporting that choline is not sensitive to this [83] and others documenting that levels of choline-containing compounds are reduced [84,86]. Some studies have looked at the effects of medication on brain choline metabolite levels [84,104]. One spectroscopic study observed reduced choline levels in the anterior cingulum following chronic methylphenidate treatment [84]. Additional research found that children with ADHD on stimulant medication had significantly higher choline ratios in the left prefrontal region, indicating that the medication could normalize brain metabolite levels [104]. Further clinical trials are needed to investigate this further. Given the advances of new medications, these should also be studied in relation to any effects on choline levels/metabolites. Another important point to consider is that sensory issues and food selectivity in these population groups could further impact dietary intakes [105,106], including that of choline.

Turning to dyslexia, more research is needed to unravel the science. It appears that higher choline levels in those with reading difficulties for reading-related regions could be reflective of higher membrane turnover, white matter, and cellular density, indicating compromised myelination in this population [87,92,93]. For dyslexia, the visual magnocellular theory posits that omega-3 long chain polyunsaturated fatty acids (particularly docosahexaenoic acid; DHA) provide membranes with properties to enable rapid electrical activity of M-cells and the rapid opening and closure of these cells [107]. Thus, a lack of DHA can affect the integrity of these neurons and result in impaired visual magnocellular function [107]. Interestingly, in the in utero environment, there is now evidence that choline is needed for the appropriate development of the visual system, especially the regulation of temporal progression of retinogenesis [42]. This is a field that would be worthy of extended study in relation to dyslexia and visual processing.

Some organizations are now beginning to specifically mention choline’s role(s) in neurodevelopment. For example, both the American Academy of Pediatrics (AAP) and the American Medical Association (AMA) have communicated the importance of maternal choline intake during pregnancy and lactation and identified that failure to provide choline during the first 1000 days post-conception could result in lifelong shortfalls in brain function, despite subsequent nutrient repletion [108,109,110,111]. The AAP calls for pediatricians to move beyond simply recommending a ‘good diet’ and ensure that pregnant women and young children have access to food that provides adequate amounts of “brain-building” nutrients, with choline being listed as one of these [111]. The main food groups reported as contributing to choline intake are milk, egg, and their derived products, as well as meat, grains and fish [112]. The AMA explains that during pregnancy, cognitive, neural tube and hippocampus development are dependent on adequate choline intake, and have called for prenatal vitamin supplements to contain ‘evidenced-based’ amounts of choline [110].

A recent analysis of over 180 commercial prenatal supplements identified that these varied in content, frequently only providing a subset of essential vitamins, and containing amounts that tended to be below recommendations [113]. The authors concluded that choline was only included in 40% of prenatal supplements, at a median level of 25 mg [113]. They also reported the incidence of certain physical and mental health conditions in the U.S.—9.4% for ADHD, and 2% for autism—and recognized that choline is needed for fetal brain development, potentially lowering the risk of neural tube defects and autism [113]. It was concluded that women who do not eat several eggs per week may benefit from prenatal supplements containing choline. Dosages provisionally advised in this publication—at least 350 mg of choline during the first two trimesters, and approximately 600 mg in the third trimester—were rather high [113]. EFSA advises an AI of 480 mg/day choline for pregnant and 520 mg/day choline for lactating women [46]. Furthermore, it is important to consider that tolerable upper intake levels have not been formally established for choline. Some side-effects, such as gastrointestinal disturbances and fishy body odor, have been reported in earlier studies administering higher amounts of choline (8–20 g/day) [114,115,116], though these studies are dated and not representative of the population of interest.

Overall, there is an emergent evidence base accruing in this important field. It is evident that more clinical trials are needed before firm conclusions can be drawn. This paper aimed to provide a first insight into the field. One prudent point to consider is the nature of the terminologies used in scientific papers published within this field. We are now gradually moving away from phrases such as ‘reading disorders’ or ‘reading disabilities’ and ‘autism spectrum disorder’ to revised terms such as ‘reading difficulties’ and ASC (autism spectrum condition). Some older terms may have been used in this paper, but only when referring to older studies that used such terms. In the future, greater consistency is needed in aligning with revised, modernized terminologies.

On a final note, now is the time to pay greater attention to choline from the perspective of neurodevelopment and NDD. Many countries, including the United Kingdom, do not yet have formal choline intake recommendations [117]. Clearly this is a central starting point. A generic lack of awareness about the nutrient choline and its potential role(s) in neurodevelopment and NDDs is evident [108,109,117]. Firming up choline recommendations and guidance to women of childbearing age potentially has tremendous implications for supporting the neurodevelopment of the next generation.

## 8. Conclusions

All taken together, choline appears to play a central role in brain development, growth, and function. An accruing body of evidence indicates that choline could have underpinning roles in the etiology of ASD, ADHD and possibly other NDDs. The origins of these conditions are multi-faceted, but can be genetic and attributed to environmental factors, including dietary exposures such as choline (in utero and beyond). Mechanisms of choline in relation to brain function and neurochemistry may be different at different life stages, e.g., in utero versus later in life, and for the variations of NDDs that exist and co-exist. Future research is needed in this important field. Choline certainly appears to be a nutrient worthy of consideration when studying neurodevelopment and NDDs.

## Figures and Tables

**Figure 1 nutrients-15-02876-f001:**
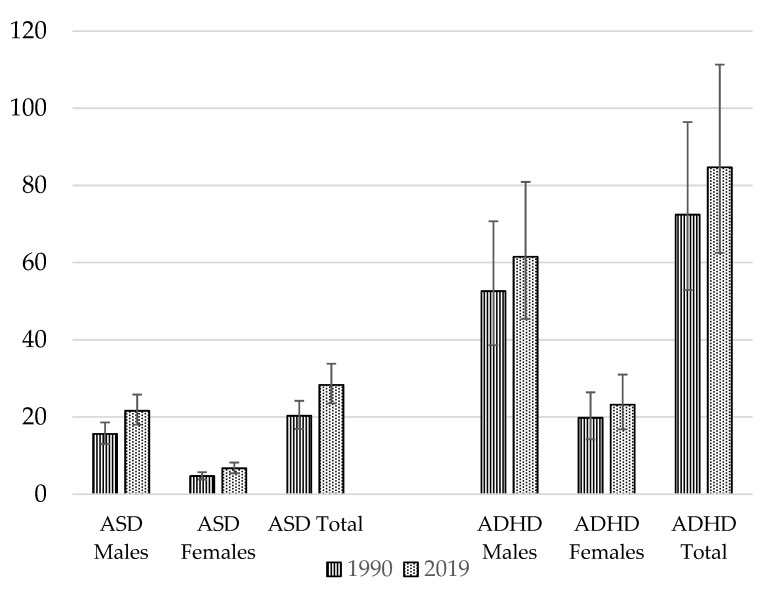
Global prevalence (in millions) of NDDs. Source: Data extracted from the Global Burden of Disease Study [8].

## Data Availability

Not applicable.

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
