# Peer review of "The Role of Choline in Neurodevelopmental Disorders—A Narrative Review Focusing on ASC, ADHD and Dyslexia"

_nutrients, 2023, doi:10.3390/nu15132876_

Round 1

Reviewer 1 Report

The present narrative review discusses the main research describing the effect of choline in neurodevelopmental disorders (NDDs), who appear to be rising in prevalence according to the recent Global Burden of Disease Study. This rise is likely to be multi-factorial, but the role of certain nutrients known to facilitate neurodevelopment should be considered. One possible contributing factor could be attributed to deficits in choline intake, particularly during key stages of neurodevelopment which includes the first 1000 days of life and childhood. Choline, a key micronutrient is crucial for an optimal offspring's neurodevelopment and brain function. This review aims to better understand its role in the aetiology and management of these. In terms of findings, low choline intakes and reduced or altered choline status have been reported in relevant population subgroups: pregnancy (in utero), children with autism spectrum disorders and attention deficit hyperactivity disorder on stimulants and those with dyslexia. The authors conclude that this an optimal choline provision may offer some neuronal protection in early life and help mitigate some of the cognitive effects attributed to neurodevelopmental conditions in later life.  Research indicates that choline may act as a modifiable risk factor for certain neurodevelopmental conditions though ongoing research is  needed.

This is an overall meaningful and intriguing review on role of choline in NDDs. The paper lists an evidence as for a need more research and also the importance of maternal choline intake during pregnancy and lactation based on existing data, hence that are recommended by both the American Academy of Pediatrics and American Medical Association. 

The only suggestion is to additionally organize the paper’s presentation. For example, by adding on 2-3 additional Tables to the body text as it appear dense under #5 Brain Imaging and Metabolite studies, and 6 Human studies. For consistency, those tables can be divided by diagnosis of ASD, ADHD, Dyslexia. Including some details of relevant clinical trials would be of particular interest for readers, which would be easier to follow if detailed in the Tables.

The authors could have combined Dyslexia and Processing speed and attention as one so called Skills (i.e., achievements-reading specific skills  and Processing speed as a cognitive category), and ADHD and ASD as another separate category termed Behavior. And keep it consistent in the body text.  First present the Skills, then the Behavior, so it is easier to follow a pattern of the valuable data presented. This, the additional Tables would add on a visual benefit easier to ‘digest’ the presented data. The authors could also further organize the body text. For example, 

For example, the data under #5 Brain Imaging and Metabolite studies also include human studies, and then # 6 are termed Human studies. Some additional effort to details here would make this paper easier to follow. 

Perhaps, starting a paragraph under the diagnoses of skills and Behaviors state a) an issue, then report on b) progress, and c) current recommendations and future directions  at the end of paragraph. In particular the ones from the American Academy of Pediatrics and American Medical Association. If the latter under c) overlaps substantially under the Skills and Behaviors, they can be combined for all three diagnoses, and some specifics emphasized, if relevant. 

No major concerns. 

Author Response

Reviewer 1

The present narrative review discusses the main research describing the effect of choline in neurodevelopmental disorders (NDDs), who appear to be rising in prevalence according to the recent Global Burden of Disease Study. This rise is likely to be multi-factorial, but the role of certain nutrients known to facilitate neurodevelopment should be considered. One possible contributing factor could be attributed to deficits in choline intake, particularly during key stages of neurodevelopment which includes the first 1000 days of life and childhood. Choline, a key micronutrient is crucial for an optimal offspring's neurodevelopment and brain function. This review aims to better understand its role in the aetiology and management of these. In terms of findings, low choline intakes and reduced or altered choline status have been reported in relevant population subgroups: pregnancy (in utero), children with autism spectrum disorders and attention deficit hyperactivity disorder on stimulants and those with dyslexia. The authors conclude that this an optimal choline provision may offer some neuronal protection in early life and help mitigate some of the cognitive effects attributed to neurodevelopmental conditions in later life.  Research indicates that choline may act as a modifiable risk factor for certain neurodevelopmental conditions though ongoing research is  needed.

Yes that is correct thank you.

This is an overall meaningful and intriguing review on role of choline in NDDs. The paper lists an evidence as for a need more research and also the importance of maternal choline intake during pregnancy and lactation based on existing data, hence that are recommended by both the American Academy of Pediatrics and American Medical Association.   Yes that is correct thank you.

The only suggestion is to additionally organize the paper’s presentation. For example, by adding on 2-3 additional Tables to the body text as it appear dense under #5 Brain Imaging and Metabolite studies, and 6 Human studies. For consistency, those tables can be divided by diagnosis of ASD, ADHD, Dyslexia. Including some details of relevant clinical trials would be of particular interest for readers, which would be easier to follow if detailed in the Tables.

As there is a table for the roles of choline in neurodevelopment and brain function, I have added extra tables for the focal points of the paper – ASD, ADHD and dyslexia. Thank you this looks good and has reinforced the paper.

The authors could have combined Dyslexia and Processing speed and attention as one so called Skills (i.e., achievements-reading specific skills  and Processing speed as a cognitive category), and ADHD and ASD as another separate category termed Behavior. And keep it consistent in the body text.  First present the Skills, then the Behavior, so it is easier to follow a pattern of the valuable data presented. This, the additional Tables would add on a visual benefit easier to ‘digest’ the presented data. The authors could also further organize the body text. For example,  

Thank you, I have -re-read through the article and further smoothed it out as you further advised below.

For example, the data under #5 Brain Imaging and Metabolite studies also include human studies, and then # 6 are termed Human studies. Some additional effort to details here would make this paper easier to follow.

Any mention of human studies in this section has been removed – it has been kept specific to animal studies.  Studies that were human have now been reallocated to the other sections and the new Tables.

Perhaps, starting a paragraph under the diagnoses of skills and Behaviors state a) an issue, then report on b) progress, and c) current recommendations and future directions  at the end of paragraph. In particular the ones from the American Academy of Pediatrics and American Medical Association. If the latter under c) overlaps substantially under the Skills and Behaviors, they can be combined for all three diagnoses, and some specifics emphasized, if relevant. Thank you, hopefully this is a smoother read now.

Reviewer 2 Report

The main goal of the study was to identify the role of choline in the development of neurodevelopmental disorders. This topic is very relevant, as the growth of neurodevelopmental disorders is growing in all countries of the world. The presented results complement the information on specific forms of neurodevelopmental disorders - ADHD, dyslexia, autism spectrum disorder. The research methodology is in general in line with the aims and objectives of the study. Tables have full material for analysis and discussion.

The conclusions are sufficiently author's vision of the presented results of the study. All references to literature have their rationale.

 However, we would like to see in this review the authors' opinions on drug therapy with choline, including multivitamin complexes (reference doi:10.1186/s40748-678022-00139-9). Recommended doses of choline-containing drugs are quite high, up to 600 mg (Line 392-395). Choline products and medicines, like any other products and medicines, have side effects on the smooth muscles of the internal organs (bladder, intestines) and allergic reactions, especially in children and pregnant women. In what cases is the use of high-dose choline-containing medications for children and pregnant women limited (prohibited) for the prevention of neurodevelopmental disorders in children?

Author Response

Reviewer 2

The main goal of the study was to identify the role of choline in the development of neurodevelopmental disorders. This topic is very relevant, as the growth of neurodevelopmental disorders is growing in all countries of the world. The presented results complement the information on specific forms of neurodevelopmental disorders - ADHD, dyslexia, autism spectrum disorder. The research methodology is in general in line with the aims and objectives of the study. Tables have full material for analysis and discussion.

Thank you for the valuable comments.

The conclusions are sufficiently author's vision of the presented results of the study. All references to literature have their rationale.

Thank you.

However, we would like to see in this review the authors' opinions on drug therapy with choline, including multivitamin complexes (reference doi:10.1186/s40748-678022-00139-9). Thank you - this reference source is mentioned but a wider critique has now been added.

Recommended doses of choline-containing drugs are quite high, up to 600 mg (Line 392-395). Choline products and medicines, like any other products and medicines, have side effects on the smooth muscles of the internal organs (bladder, intestines) and allergic reactions, especially in children and pregnant women. In what cases is the use of high-dose choline-containing medications for children and pregnant women limited (prohibited) for the prevention of neurodevelopmental disorders in children?  Thank you – I have edited this part.  As you mention this is quite high and it is probably somewhat premature to stay this.  I have mentioned EFSA Dietary Reference Values

https://www.efsa.europa.eu/en/efsajournal/pub/4484

and commented on the lack of a Tolerable Upper Limit and potential excess ramifications

https://efsa.onlinelibrary.wiley.com/doi/epdf/10.2903/j.efsa.2016.4484 pg 10.